



# The sea ice model component of HadGEM3-GC3.1

Jeff K. Ridley[1], Edward W. Blockley[1], Ann B. Keen[1], Jamie G.L. Rae[1], Alex E. West[1], David Schroeder[2]

[1]Met Office, Exeter, EX1 3PB, UK
[2]Department of Meteorology, University of Reading, Reading, RG6 6BB, UK

*Correspondence to*: Jeff K. Ridley (jeff.ridley@metoffice.gov.uk)

**Abstract.** A new sea ice configuration, GSI8.1, is implemented in the Met Office global coupled configuration HadGEM3-GC3.1 which will be used for all CMIP6 simulations. The coupling between atmosphere and sea ice has been improved to increase stability in the thermodynamic solver. Here we describe the sea ice model component and show that the Arctic
thickness and extent compare well with observationally based data.

## 1. Introduction

HadGEM3-GC3.1 is the global coupled model configuration to be submitted for simulations to CMIP6 by the Met Office Hadley Centre (Williams et al., in press). It is comprised of global atmosphere, GA7.1 and land surface GL7.0 components (Walters et al., in prep), coupled to global ocean GO6 (Storkey et al., in prep) and global sea ice GSI8.1 components. This
paper describes the global sea ice component (GSI8.1) which is embedded in the NEMO (Madec, 2008) ocean configuration (GO6) and uses a tripolar grid, while the atmosphere model (GA7.1) and land surface (GL7.0) a configuration of JULES (Best et al., 2011), use a staggered latitude-longitude grid. The communication between the two components is through the OASIS coupler (Valcke, 2006).

The model resolutions of HadGEM3-GC3.1 (hereafter referred to as GC3.1) to be submitted to CMIP6 are N96 atmosphere
(135km in midlatitudes) with 1 degree ocean (ORCA1), N216 atmosphere (60km in midlatitudes) with 1/4 degree ocean (ORCA025) and N512 atmosphere (25km in midlatitudes) with 1/12 degree ocean (ORCA12). For the purposes of model evaluation we shall present results from the N216-ORCA025 resolution model.

## 2. Model description

The GSI8.1 Global Sea Ice configuration builds on the previous version GSI6 (Rae et al., 2015) and is based on version 5.1
the Los Alamos sea ice model CICE (Hunke et al., 2015). The CICE model determines the spatial and temporal evolution of the ice thickness distribution (ITD) due to advection, thermodynamic growth and melt, and mechanical redistribution/ridging. At each model grid point the sub-grid-scale ITD is modelled by dividing the ice pack into five thickness categories, with an additional ice-free category for open water areas. The initial implementation of CICE within

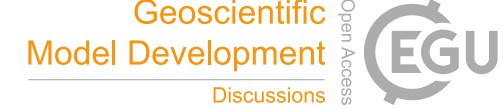



the HadGEM3 coupled climate model is described in Hewitt et al. (2011). Appendix 1 contains details of namelist options and parameters used in GSI8.1 as well as C pre-processing keys used to build the model.

### 2.1 Albedo scheme

The albedo scheme used in this configuration is based on the scheme used in the CCSM3 model (see Hunke et al., 2015), and has separate albedos for visible (< 700 nm) and near-infrared (> 700 nm) wavelengths for both bare ice and snow. The scheme is described in Section 3.6.2 of the CICE User's Manual (Hunke et al., 2015). Penetration of radiation into the ice, as described by Hunke et al. (2015), is not included here. For this reason, following Semtner (1976), a correction is applied to the surface albedo to account for scattering within the ice pack.

This configuration includes the impact of surface melt ponds on albedo as an addition to the CCSM3 albedo scheme. The melt pond area fraction, $f_p(n)$, and depth, $h_p(n)$, for ice in thickness category $n$, are calculated with the CICE topographic melt pond formulation (Flocco et al., 2010, 2012; Hunke et al., 2015). Where the pond depth, $h_p(n)$, on ice of thickness category $n$ is shallower than 4 mm, the ponds are assumed to have no impact on albedo, and the albedo, $\alpha_{pi}(n)$, of such ponded ice is simply equal to that of bare ice, $\alpha_i$. Where the pond depth is greater than 20cm, the underlying bare ice is assumed to have no impact, and the ponded ice albedo is assumed to be equal to that of the melt pond, $\alpha_p$. For ponds deeper than 4 mm but shallower than 20 cm, the underlying bare ice is assumed to have an impact on the total pond albedo, and the bare ice and melt pond albedos are combined linearly:

$$\alpha_{pi}(n) = \frac{h_p(n)}{0.2}\,\alpha_p + \left(1 - \frac{h_p(n)}{0.2}\right)\alpha_i.$$

Because the impact of melt ponds on albedo has been included explicitly, the reduction in bare ice albedo with increasing temperature (Hunke et al., 2015), which was intended to account for melt pond formation, is not included. However, the reduction in snow albedo, $\alpha_s(n)$, with increasing surface skin temperature, intended to take account of the lower albedo of melting snow, has been retained, and takes the form:

$$\alpha_s(n) = \begin{cases} \alpha_c & \text{if } T(n)<T_c \\ \alpha_c + \left(\frac{\alpha_m - \alpha_c}{T_m - T_c}\right)(T(n) - T_c) & \text{if } T(n)\geq T_c \end{cases},$$

where $\alpha_c$ and $\alpha_m$ are the albedos of cold and melting snow respectively, $T_m$ is the snow melting temperature (i.e., 0°C), $T(n)$ is the surface skin temperature of ice in thickness category $n$, and $T_c$ is some threshold temperature at which melting starts to affect the snow albedo.

The scheme calculates the total gridbox albedo, $\alpha(n)$, of ice in thickness category $n$, for each of the two wavebands by combining the albedo, $\alpha_{pi}(n)$, of the ponded fraction, calculated as described here, with the albedos of bare ice, $\alpha_i$, and snow, $\alpha_s(n)$, weighted by the melt pond fraction, $f_p(n)$, and the snow fraction, $f_s(n)$:

$$\alpha(n) = f_p(n)\alpha_{pi}(n) + \left(1 - f_p(n)\right)(f_s(n)\alpha_s(n) + (1 - f_s(n))\alpha_i).$$

The snow fraction, $f_s(n)$, for category $n$, is parameterised via a calculation based on snow depth, $h_s(n)$:

$$f_s(n) = \frac{h_s(n)}{h_s(n) + h_{snowpatch}},$$



where $h_{snowpatch}$ is a length scale parameter (Hunke et al. 2015). Note that this is different from the parameterisation used in the previous configuration, GSI6.0, described by Rae et al. (2015).

## 2.2. Thermodynamics

GSI8.1 is the first sea ice configuration of the Met Office model to use multi-layer thermodynamics. Previously, the sea ice

model used the zero-layer formulation described in the appendix to Semtner (1976), in which surface temperature reacts instantaneously to surface forcing, and conduction within the ice is uniform. In the new formulation, the sea ice has a heat capacity, and hence conduction can vary in the vertical. The ice is divided into four vertical layers, each with its own temperature and prescribed salinity; an additional snow layer is permissible on top of the ice (Figure 1). The thermodynamics scheme is very similar to that described by Bitz and Lipscomb (1999), present in CICE5.1.2, in which the

diffusion equation with temperature-dependent coefficients is solved by the iteration of a tridiagonal matrix equation. However, it is modified as described by West et al (2016), with surface exchange calculations carried out in the Met Office surface exchange scheme, JULES. The diffusion equation is forced from above by the conductive flux from the ice surface into the top layer interior, a variable calculated by the surface exchange and passed to the ice model; the top layer temperature, thickness and conductivity become the bottom boundary condition for the next iteration of the surface

exchange.

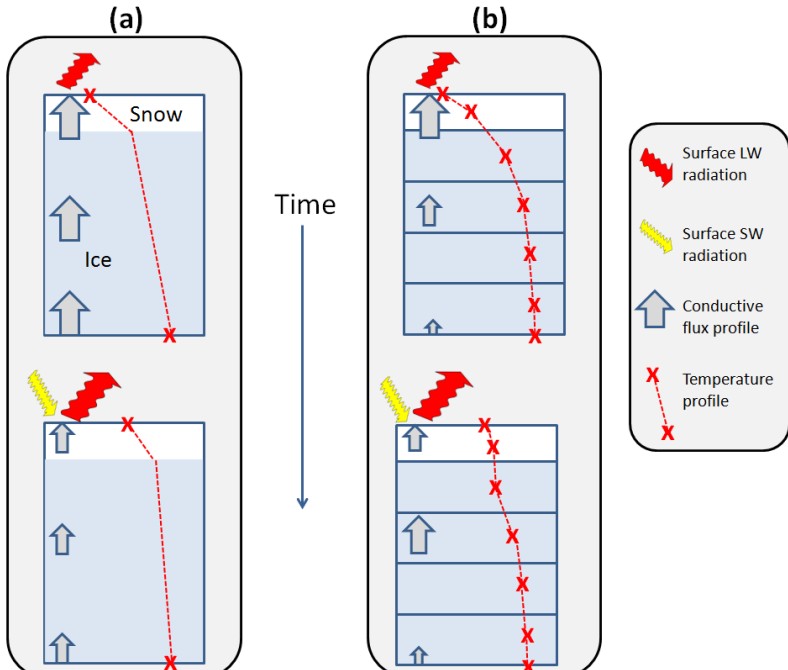

**Figure 1. Schematic demonstrating the evolution of: (a) the old, zero-layer ice thermodynamics scheme; (b) the new, multilayer scheme, under the application of a surface SW forcing.**





### 2.2.1 Semi-implicit coupling

The application of the above coupling method to three-dimensional modelling, in which the ocean and sea ice are on different grids to the atmosphere, initially caused some problems related to the apportionment of energy fluxes between ice cells. The OASIS-MCT coupler used by the Met Office does not have the functionality to regrid variables with time-varying

weights; hence the conductive flux $F_{condtop}$, calculated by the surface exchange in a single atmosphere grid cell, was divided amongst the underlying ocean model cells in proportion to grid cell area, regardless of the fraction of ice present in those cells. In practice, this meant that cells with a low ice fraction received too much energy, while cells with a high ice fraction received too little. In a large number of instances this resulted in the CICE temperature solver being forced with exceptionally high local conductive fluxes, rendering convergence of the iterative solver difficult or impossible.

In order to deal with this problem, and render the coupling more physically realistic, the coupling was made semi-implicit, making use of the fact that in GC3.1 ocean-atmosphere variables are passed through the coupler before atmosphere-ocean variables. In the new method, the sea ice fraction was passed by first-order conservative regridding to the atmosphere at a coupling instant, and this new sea ice fraction was used in JULES to divide through $F_{condtop}$ to produce a 'pseudo-local' conductive flux. This new flux was then passed to the ocean model in the normal way, where it was multiplied by the same

ice fraction field on the ocean grid to produce the grid-box-mean field that would be used for the rest of the time step (Figure 2). This alternative grid-box-mean field has the properties of conserving energy, and of fluxes being of a magnitude roughly proportional to the underlying ice fraction, resulting in a more homogeneous, realistic local flux field with which to force the CICE temperature solver. In fact this field is exactly equivalent to the physically desirable solution that would be produced if fluxes were divided amongst underlying ocean grid cells in proportion to ice area (not shown).





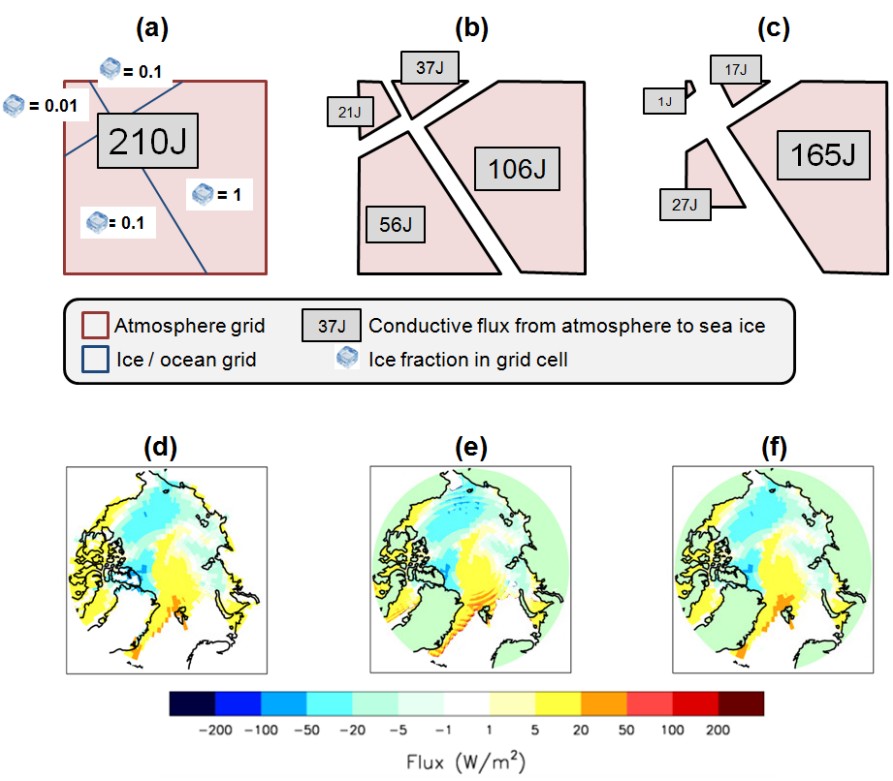

**Figure 2. Demonstrating the implementation, and effect, of semi-implicit coupling: (a) an example atmospheric grid cell, overlying 4 ice/ocean grid cells calculates a sea ice conductive flux representing 210J energy; (b) the division of energy between cells with standard coupling, in proportion to grid cell area; (c) the division with semi-implicit coupling, in proportion to ice area. (d) shows a conductive flux field on the atmosphere grid; the resulting flux field on the sea ice grid is shown for (e) standard coupling and (f) semi-implicit coupling.**

Even after implementation of the semi-implicit coupling there remained two cases in which the CICE temperature solver, forced by conductive flux, would fail to converge. In the first case, convergence becomes very slow for thin, melting ice; particularly in summer, the surface exchange scheme would occasionally calculate large conductive fluxes for which the solver would not converge within the required 100 iterations. To deal with this problem, a maximum threshold of $1000h_I$ was specified for the conductive flux, where $h_I$ is ice thickness; any surplus conductive flux above this value would be redirected to the bottom of the ice, and added to the ice-to-ocean heat flux.

The second problem was subtler, and caused by an instability related to the way in which the CICE thickness distribution interacts with the coupling method. In the Arctic, it is common for a large number of ocean cells each to underlie partially a single atmospheric grid cell. In cold winter conditions, conducive to strong ice growth, the fraction of ice in the first (thinnest) category, $a_1$, usually becomes quite small. With cold atmospheric temperatures, surface flux and conduction through ice in this category are necessarily strongly upwards; in some cells, random effects, perhaps dynamical, would cause



conduction to be stronger than in others, and also lead to lower top layer temperatures. However, stronger conduction also promotes ice growth, which reduces the fraction $a_1$ in the grid cell, as it gets promoted to $a_2$, rendering its top layer temperature less visible to the atmosphere. In a small number of cases this was found to cause runaway cooling, with top layer temperature in isolated cells cooling to below -100°C, forced by high negative conductive fluxes calculated by a

surface exchange scheme that was seeing much higher gridbox mean ice temperatures. This problem was solved by linearly reducing conductive flux to zero as top layer ice temperature fell from -60°C to -100°C, with the excess flux passed directly to the bottom of the ice (and therefore helping to grow more ice, the effect that would be expected to occur in reality).

## 2.3. Dynamics

The standard elastic-viscous-plastic rheology (EVP) for ice dynamics in CICE is used here (Hunke et al., 2015). However,

NEMO is on a C-grid and CICE on a B-grid. This is dealt with though simple interpolation from CICE to NEMO as described in Hewitt et al. (2011).

## 3. Model evaluation

An example of the sea ice evaluation provided here is the Arctic multiannual mean winter (December, January and February, DJF) ice thickness as diagnosed by the model (Figure 3), both in its CMIP6 configuration of GC3.1 and its previous stable

version GC2 (Williams et al., 2015). The model present-day control is forced by greenhouse gases and aerosols from year 2000 for 100 simulated years. The evaluation data is Cryosat-2 satellite thickness (Tilling et al., 2016) inferred from freeboard measurements from 2011-2015 along with the 1990-2010 mean thickness from Pan-Arctic Ice Ocean Modelling and Assimilation System sea ice reanalysis (PIOMAS; Schweiger et al., 2011), inferred through the assimilation of observed sea ice fraction. The Cryosat-2 thickness retrievals are included solely as a guide to the ice thickness distribution as the ice

has thinned since 2000. Figure 3 also show the model sea ice extent, depicted by the 15% ice concentration contour, compared with the 1990-2009 mean from the HadISST1.2 sea ice analysis (Rayner et al., 2003). The PIOMAS and GC3.1 Arctic ice thicknesses are comparable in spatial pattern save for PIOMAS depicting a larger area of thick ice adjacent to North Greenland and the Canadian Archipelago and GC3.1 depicting thicker ice in the central Arctic. CryoSat-2 depicts thinner ice in the western Arctic. Both GC3.1 and CryoSat-2 show thicker ice along the east Greenland coast than PIOMAS.

The DJF ice extent in GC2 is low compared with the PIOMAS analysis and CryoSat-2 data, but consistent with the low ice thickness. The extent compares well with the HadISST analysis in GC3.1, however, the ice is perhaps (this being thin ice that microwave observations, the basis of HadISST, may not be able detect) overly extensive in the Greenland and Norwegian seas. The winter sea ice extent simulated by GC3.1 is much closer to the HadISST observations than was the case for GC2 in the Bering/Chukchi, Barents and Labrador Seas.

Figure 4 shows the mean seasonal cycle of volume for the GC3.1 model compared with that from GC2. The veracity of the model seasonal cycle of sea ice volume informs us if the annual energy budget to the ice is well balanced. Unfortunately





there are few observational means to assess this and so here we use the PIOMAS model as a reference in the Arctic, and satellite estimates from ICESat for the Antarctic (Kurtz & Markus, 2012). It can be seen that, in agreement with Figure 3, the GC2 Arctic ice volume is low, and that it is in closer agreement with PIOMAS at GC3.1. Both models have near identical annual cycles, but offset, indicting the atmospheric and oceanic forcing of the Arctic sea ice is largely unchanged between

5    the two model versions. The estimates for 2003-2008 Antarctic ice volume from ICESat are for a minimum of 3357 km$^3$ in summer to a maximum of 11,111 km$^3$ in winter (Figure 4). The GC3.1 volume compares well with ICESat in the summer (2735 km$^3$) but is a little too high in the winter (17,087 km$^3$). The GC2 configuration had a warm bias in the Southern Ocean (Williams et al., in press) which was melting the Antarctic ice. This bias has been considerably reduced in CG3.1 (Hyder et al., under review) resulting in a substantial increase in the Antarctic sea ice volume.

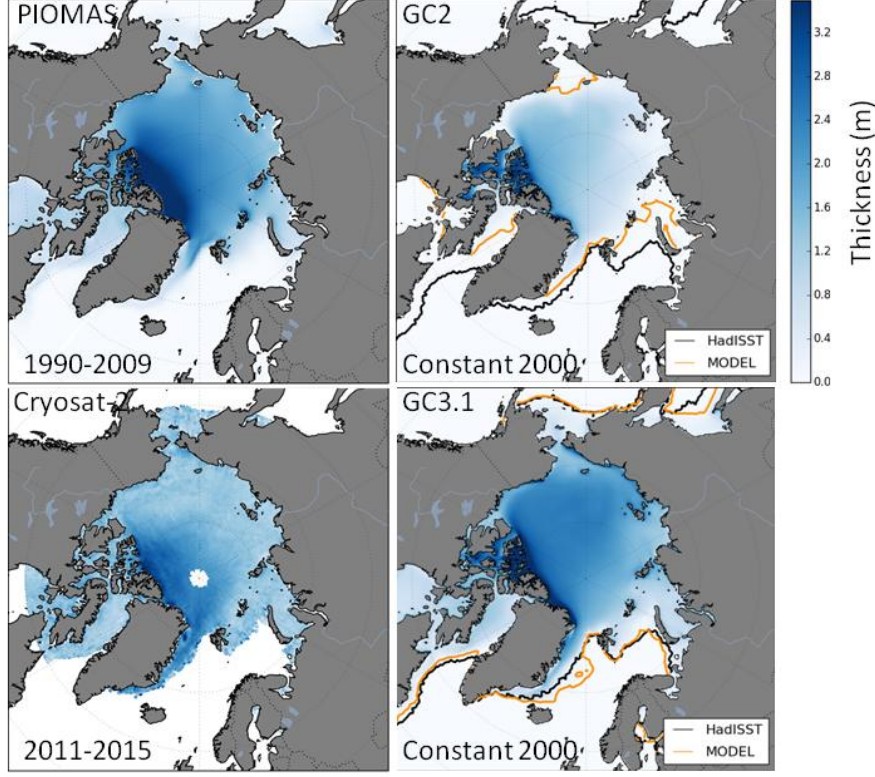

**Figure 3. The HadGEM3 model thickness (right) compared with that from the PIOMAS model reanalysis (1990-2009) and thickness inferred from the CryoSat-2 (2011-2015) sea ice freeboard measurements (left). The orange and black lines show the 15% ice concentration contours for the model simulations and the HadISST1.2 sea ice analysis respectively.**

15





**Figure 4. The HadGEM3 model annual cycle of sea ice volume in the Arctic (upper) and Antarctic (lower). Volume estimates from the PIOMAS model reanalysis are included for the Arctic and ICESat estimates of volume in the Antarctic (grey dashed lines).**

## 4. Summary

The GSI8.1 sea ice configuration of the Met Office Hadley Centre CMIP6 coupled model HadGEM3-GC3.1 has a number of physical enhancements compared to the previous version GSI6, including the introduction of multilayer thermodynamics and an explicit representation of the radiative impact of meltponds. A semi-implicit coupling scheme refines the transpose of atmospheric fluxes to the sea ice, improving the stability of the thermodynamic solver. The final GC3.1 namelist options and pre-processor keys (see Tables 1 and 2 in the Appendix) produce ice thickness and extent that are in good agreement with analyses.



**Acknowledgements**

This work was supported by the Joint UK BEIS/Defra Met Office Hadley Centre Climate Programme (GA01101). The CryoSat-2 sea ice thickness data was provided by the Centre for Polar Observation and Modelling (CPOM), supported by the Natural Environment Research Council (cpom300001).

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

**Appendix**

**Table 1: GSI8.1 namelist options and hard-wired parameters used within CICE and JULES**

| CICE namelists | | |
|---|---|---|
| [namelist:dynamics_nml] | | |
| | advection='remap' | Remapping as a transport/advection algorithm |
| | kdyn=1 | Using EVP rheology for the dynamics |
| | revised_evp=.false. | Standard CICE EVP formulation |
| | krdg_partic=1 | Exponential dependence on ice strength for ridging participation |



| | | function |
| --- | --- | --- |
| | krdg_redist=1 | Exponential ITD redistribution function for ridging |
| | kstrength=1 | Rothrock (1975) formulation for ice strength |
| | mu_rdg=3.0 | e-folding scale of ridged ice ($m^{0.5}$) |
| | ndte=120 | Number of time steps for internal stress calculations |
| [namelist:ponds_nml] | | |
| | hp1=0.01 | Critical pond lid thickness (m) |
| | rfracmax=0.85 | Maximum retained fraction of melt-water |
| | rfracmin=0.15 | Minimum retained fraction of melt-water |
| [namelist:thermo_nml] | | |
| | ktherm=1 | Multilayer thermodynamics (0=zero layer, 2=mushy layer) |
| | saltmax=9.6 (NB. normally hard-wired in the CICE code) | Maximum salinity at the ice-ocean interface for calculation of fixed salinity profile |
| JULES namelist | | |
| | nice_use=5 | Number of sea ice thickness categories used in surface exchange |
| | albicev_cice=0.78 | Visible albedo of bare ice |
| | albicei_cice =0.36 | Near-IR albedo of bare ice |
| | albsnowv_cice =0.98 | Visible albedo of cold snow |
| | albsnowi_cice =0.70 | Near-IR albedo of cold snow |
| | emis_sice= 0.9760 | emissivity of sea ice |
| | albpondv_cice =0.27 | Visible albedo of  melt ponds |





| | | |
|---|---|---|
| | albpondi_cice =0.07 | Near-IR albedo of melt ponds |
| | dalb_mlts_v_cice =-0.10 | Change in snow Visible albedo per degree C rise in temperature |
| | dalb_mlts_i_cice =-0.15 | Change in snow Near-IR albedo per degree C rise in temperature |
| | dt_snow_cice =1.0 | Permitted range of snow temperature over which albedo changes (K) |
| | ahmax=0.3 | Sea ice thickness (m) below which albedo is influenced by underlying ocean |
| | pen_rad_frac_cice =0.4 | Semtner correction: fraction of SW radiation that penetrates sea ice and scatters back |
| | sw_beta_cice= 0.6 | Semtner correction: attenuation parameter for SW in sea ice which controls the additional albedo due to internal scattering |
| | snowpatch=0.02 | Length scale for parameterisation of non-uniform snow coverage (m) |
| | z0miz=0.1 | Roughness length for MIZ (m) |
| | z0sice=0.0005 | Roughness length for pack ice (m) |
| | z0h_z0m_miz=0.2 | Ratio of thermal to momentum roughness lengths for marginal ice |
| | z0h_z0m_sice=0.2 | Ratio of thermal to momentum roughness lengths for pack ice |
| Hard-wired parameters | | |
| | kice =2.03 | Thermal conductivity of fresh ice(W/m/K) |
| | ksno = 0.31 | Thermal conductivity of snow |



| | rhos=330.0 | Density of snow (kg/m$^3$) |
|---|---|---|
| | | (W/m/K) |
| | dragio=0.01 | Ice – ocean drag coefficient |

**Table 2: C preprocessor keys used to build the GSI8.1 CICE component of HadGEM3-GC3.1**

| CPP key | Purpose |
|---|---|
| LINUX | Building CICE for the Linux environment |
| ncdf | NetCDF format options available for input and output files |
| CICE_IN_NEMO; key_nemocice_decomp | CICE is run within the NEMO model on the same processor decomposition |
| ORCA_GRID | Using the ORCA family of grids |
| coupled; key_oasis3mct; key_iomput | Coupled model run passing variables through NEMO and using the OASIS3 MCT coupler |
| REPRODUCIBLE | Ensures global sums bit compare for parallel model runs with different grid decompositions |
| gather_scatter_barrier | Use MPI barrier for safer gather and scatter communications |
| NICECAT=5; NICELYR=4; NSNWLYR=1 | 5 thickness categories, 4 ice layers, 1 snow layer |
| TRAGE=1; TRPND=1 | Using single ice age and melt-pond tracers |