# Peer review of "The sea ice model component of HadGEM3-GC3.1"

_Geoscientific Model Development, 2017_

## Short Comment (SC1) · 4 Oct 2017

Dear authors,

In my role as Executive editor of GMD, I would like to bring to your attention our Editorial version 1.1:

http://www.geosci-model-dev.net/8/3487/2015/gmd-8-3487-2015.html

This highlights some requirements of papers published in GMD, which is also available on the GMD website in the 'Manuscript Types' section:

http://www.geoscientific-model-development.net/submission/manuscript_types.html

In particular, please note that for your paper, the following requirements have not been

met in the Discussions paper:

- "All papers must include a section, at the end of the paper, entitled 'Code availability'. Here, either instructions for obtaining the code, or the reasons why the code is not available should be clearly stated. It is preferred for the code to be uploaded as a supplement or to be made available at a data repository with an associated DOI (digital object identifier) for the exact model version described in the paper. Alternatively, for established models, there may be an existing means of accessing the code through a particular system. In this case, there must exist a means of permanently accessing the precise model version described in the paper. In some cases, authors may prefer to put models on their own website, or to act as a point of contact for obtaining the code. Given the impermanence of websites and email addresses, this is not encouraged, and authors should consider improving the availability with a more permanent arrangement. After the paper is accepted the model archive should be updated to include a link to the GMD paper."

- Inclusion of Code and/or data availability sections is mandatory for all papers and should be located at the end of the article, after the conclusions, and before any appendices or acknowledgments. For more details refer to the code and data policy.

Please include a code availability section upon the revision of your article and provide it apriori as discussion answer to my comment.

Yours,

Astrid Kerkweg
* * *

---

## Author Comment (AC1) · 6 Oct 2017

A section on code and data availability will be included in the reviewed version of the paper

---

## Referee Comment (RC1) · Anonymous Referee #1 · 17 Oct 2017

The manuscript describes the sea-ice component GSI8.1 of the Met Office coupled configuration HadGEM3-GC3 and its evaluation against PIOMASS reanalysis and Cryosat satellite measurements. This will certainly be a reference paper for CMIP6 simulations and therefore I recommend its publication but with some substantial modifications.

General comments:

1) It is very unclear what are the novelties in this version of the sea-ice component (GSI8) compared to the one used for CMIP5 (GSI6). I recommend to explain it in the introduction or at the beginning of section 2.

2) It would be valuable to know why the most recent developments in CICE are not

included in GSI8, especially those which have been available for a couple of years. I am thinking of Delta-Eddington scheme, more elaborate melt ponds (which maybe necessitates the use of D-E), EAP rheology, mushy layers etc. Is it a question of timing, performance, robustness (testing)?

3) Besides novelties that need to be explained in details, I think the authors should not forget to give information about the rest of the ice code. Is there an interactive salinity, a fixed profile or a constant salinity? What scheme is used for advection (remapping)? And ridging/rafting? Do they consider biogeochemistry in the ice?

4) Description of JULES coupling lacks clarity but I come back to it in the "specific comments" section

5) Did the authors evaluate conservation of mass, salt and heat in the system? What are the leaks (if any) and where do they come from?

6) What is the performance of GC3 compared to GC2 in terms of CPU since new physics have been added? How much slower it is?

Specific comments:

— 2.1 albedo —

l.6: I do not understand why penetration of radiation is not included. Is it related to JULES? If yes, the authors should say it.

l. 12-15: I understand that the impact of melt ponds on albedo is linear with pond depth in the range [4 mm - 20 cm]. I would have thought more about an exponential dependency as in Lecomte et al. (2011, 2015). Is there a reference for the linear dependency?

l. 23: I suppose the value of Tc is -1degC as in Hunke et al. (2015), then refers to dt_snow_cice in the namelist? Maybe it should be stated more clearly. More generally it is sometimes difficult to relate the namelist variables with the text.

l. 30: What does Hsnowpatch represent? Snow blowing by winds? But then, where does this value (2 cm?) come from?

— 2.2 Thermodynamics —

I find that the JULES interface is poorly described here.

For example, it is said in the text that the ice model passes to JULES the top layer temperature, thickness and conductivity. Well it would be clearer to state all the variables that are exchanged: ice concentration, ice and snow thicknesses, temperature and conductivity of the first layer, concentration and thickness of ponds (I think that's all?)

Also, the authors should say that with JULES there is no transmitted radiation in and through the ice (from what I understood). What consequences do the authors expect on ice temperature or ocean heat budget?

Moreover, the reader must understand why JULES has been chosen (in just a couple sentences).

l. 6: precise that heat capacity depends on temperature and salinity.

l. 8: Fig 1 is not properly described (neither in the text nor in the caption). "Old" means the formulation used in GSI6?

— 2.2.1 semi-implicit coupling —

p.4, l. 10-18: this part is kind of unclear. I would suggest to rewrite it in a simpler way like: Problem to solve = a) non convergence of the temperature solver, b) coupling not physical. Solution = sea-ice fraction passed to the atm. which allows a semi-implicit calculation of the splitting of Fcond onto the ocean grid cells. This flux is then multiplied by ice fraction in each cell. . .

p.5, l. 11: "1000Hi". What does this threshold mean?

p.5, l. 12: Is there any quantification of this "surplus conductive flux"?

I guess there is still a small heat leak due to the non-perfect convergence of the iterative procedure. I think it would be valuable to diagnose it. In Hunke et al. (2015), it is said that this flux is about 0.01 W/m2. Does this value still hold? What does it mean in terms of global heat leak?

— 3. model evaluation —

p.6 l. 27: "this being thin ice that microwave...able detect". I do not understand this sentence.

p.7 l. 3-5: Do the authors mean that changes in Arctic sea-ice can be solely attributed to changes in the sea-ice component (contrary to the Antarctic)? I am not convinced that identical seasonal cycles indicate that forcings are unchanged. The mean forcing could also change while keeping the same annual cycle. Am I wrong?

p.7 l. 9: Concerning the great improvement in Antarctic sea ice (at least in summer), I understand it is solely due to the reduction of the warm bias in the ocean but does the atmosphere play any role here?

— References —

Lecomte, O., Fichefet, T., Vancoppenolle, M., & Nicolaus, M. (2011). A new snow thermodynamic scheme for large-scale sea-ice models. Annals of Glaciology, 52(57), 337-346.

Lecomte, O., Fichefet, T., Flocco, D., Schroeder, D., & Vancoppenolle, M. (2015). Interactions between wind-blown snow redistribution and melt ponds in a coupled ocean–sea ice model. Ocean Modelling, 87, 67-80.

---

## Referee Comment (RC2) · Anonymous Referee #2 · 27 Nov 2017

The authors describe the new sea ice model component of the climate model HadGEM3-GC3.1. As this model will be applied in a large number of experiments, this description is an important source of information for the community and will be very helpful for the users of the outputs of HadGEM3-GC3.1. The paper is short but focused on the important points. However, I would personally have been interested in a comparison of the choices performed here with the ones made in other state-of-the-art models, not just in the previous or other versions of the same model. This would have provided a wider perspective of the paper.

For me, the only point that absolutely requires additional discussion is the coupling and how it interacts with the ice thickness distribution scheme. If I understand well (page 3, lines 11-15), the surface temperature of the sea ice or snow and surface

melt are computed by the land surface module that transfers to the sea ice model a conduction flux which is used as boundary condition for the sea-ice model. This is described in West et al. (2016) but a few additional explanations would be helpful here in the revised version. The distribution of the conductive flux between the grid cells is described in section 2.2.1. However, I was not able to understand from the information given in the manuscript the method applied to distribute this conductive flux among ice thickness categories. It is also not clear to me if the surface melt is the same for all the categories. The thickness is mentioned page 5, lines 11-13 but it is to solve a specific problem of instability, not related to the ice thickness distribution. It is expected that the melting and conduction fluxes are very different between the different categories as they should be strongly related to the thickness. If in the proposed implementation, the same conductive flux is given for all categories, my interpretation is that nearly all the interest of the ice thickness distribution scheme is lost. If this is the case, this should be clearly mentioned as a strong limitation of the approach. If I have misunderstood something, the authors should explain their arguments justifying their choices and how this take advantage of the ice thickness distribution scheme.

I have also some very minor remarks. 1/ Does the atmospheric component have a specific name (as the ocean or surface ones) ? 2/ Page 11, line 11. This would be useful to list the 'GC3.1 ocean-atmosphere variables'. 3/ Page 6, lines 28-29. Here and for the other points mentioned in the manuscript, this would be interesting to discuss briefly the causes of the differences of the results of GC2 and GC3.1. 4/ Page 7, lines 4-5. I do not understand why similar annual cycles but with an offset indicate similar oceanic and atmospheric forcing. I guess a change in oceanic or atmospheric forcing could also be the origin of the offset. 5/ Page 7, line 5. A comparison with the observed ice extent in the Southern Ocean should also be provided. Figure 3. The months selected should be given in the figure caption.
* * *

---

## Author Response (AR1)

**The sea ice model component of HadGEM3-GC3.1**

Jeff K. Ridley[1], Edward W. Blockley[1], Ann B. Keen[1], Jamie G.L. Rae[1], Alex E. West[1], David Schroeder[2]

[1]Met Office, Exeter, EX1 3PB, UK
[2]Department of Meteorology, University of Reading, Reading, RG6 6BB, UK

*Correspondence to*: Jeff K. Ridley (jeff.ridley@metoffice.gov.uk)

**Abstract.** A new sea ice configuration, GSI8.1, is implemented in the Met Office global coupled configuration HadGEM3-GC3.1 which will be used for all CMIP6 simulations. The inclusion of multi-layer thermodynamics has required a semi-implicit coupling scheme between atmosphere and sea ice to ensure stability of the solver Here we describe the sea ice model component and show that the Arctic thickness and extent compare well with observationally based data.

**1. Introduction**

HadGEM3-GC3.1 is the global coupled model configuration to be submitted for physical model simulations to CMIP6 by the Met Office Hadley Centre (Williams et al., 2017). It is comprised of global atmosphere, GA7.1 and land surface GL7.0 components (Walters et al., in prep), coupled to global ocean GO6 (Storkey et al., in prep) and global sea ice GSI8.1 components. This paper describes the global sea ice component (GSI8.1) which is embedded in the NEMO (Madec, 2008) ocean configuration (GO6) and uses a tripolar grid, while the atmosphere model (GA7.1), a configuration of the Met Office Unified Model (MetUM), and land surface (GL7.0) a configuration of JULES (Best et al., 2011), use a staggered latitude-longitude grid. The communication between the two components is through the OASIS-MCT coupler (Valcke, 2006).

The model resolutions of HadGEM3-GC3.1 (hereafter referred to as GC3.1) to be submitted to CMIP6 are N96 atmosphere (135 km in midlatitudes) with 1 degree ocean (ORCA1), N216 atmosphere (60 km in midlatitudes) with 1/4 degree ocean (ORCA025) and N512 atmosphere (25km in midlatitudes) with 1/12 degree ocean (ORCA12). For the purposes of model evaluation we shall present results from the N216-ORCA025 resolution model.

**2. Model description**

Relative to its predecessor GC2, GC3.1 has new modal aerosol and multi-layer snow schemes, a newly introduced multi-layer sea ice scheme (described below) and a number of parametrization changes in all the model components, including a set relating cloud and radiation and revision to the numerics of convection (Williams et al., 2017). The GSI8.1 Global Sea Ice configuration builds on the previous version used in HadGEM3-GC2.0, GSI6 (Rae et al., 2015), and is based on version 5.1.2 of the Los Alamos sea ice model CICE (Hunke et al., 2015). The CICE model determines the spatial and temporal evolution of the ice thickness distribution (ITD) due to advection, thermodynamic growth and melt, and mechanical

**Comment [j1]:** R1-1

redistribution/ridging. At each model grid point the sub-grid-scale ITD is modelled by dividing the ice pack into five thickness categories, with an additional ice-free category for open water areas. The initial implementation of CICE within the HadGEM3 coupled climate model is described in Hewitt et al. (2011). The key differences between GSI6 and GSI8.1 are the replacement of zero-layer thermodynamics with a multi-layer scheme, the addition of prognostic melt ponds, and the

5 coupling to the atmosphere on ice thickness categories. For GSI8.1 the ice-atmosphere coupling is undertaken by category with all thermodynamic fluxes (conduction, surface melt and sublimation), as well as snow depth and melt pond fraction and depth, being calculated separately for each ice thickness category (ITC). Appendix A contains details of namelist options and parameters used in GSI8.1 and Appendix B the C pre-processing keys used to build the model.

**2.1 Albedo scheme**

10 The albedo scheme used in GSI8.1 is based on the scheme used in the CCSM3 model (see Hunke et al., 2015), and has separate albedos for visible (< 700 nm) and near-infrared (> 700 nm) wavelengths for both bare ice and snow. The scheme is described in Section 3.6.2 of the CICE User's Manual (Hunke et al., 2015). Penetration of radiation into the ice, as described by Hunke et al. (2015), is not included here. For this reason, following Semtner (1976), a correction is applied to the surface albedo to account for scattering within the ice pack.

15 This configuration includes the impact of surface melt ponds on albedo as an addition to the CCSM3 albedo scheme. The melt pond area fraction, $f_p(n)$, and depth, $h_p(n)$, for ice in thickness category $n$, are calculated with the CICE topographic melt pond formulation (Flocco et al., 2010, 2012; Hunke et al., 2015). Where the pond depth, $h_p(n)$, on ice of thickness category $n$ is shallower than 4 mm, the ponds are assumed to have no impact on albedo, and the albedo, $\alpha_{pi}(n)$, of such ponded ice is simply equal to that of bare ice, $\alpha_i$. Where the pond depth is greater than 20cm, the underlying bare ice is assumed to have no

20 impact, and the ponded ice albedo is assumed equal to that of the melt pond, $\alpha_p$. For ponds deeper than 4 mm but shallower than 20 cm, the underlying bare ice is assumed to have an impact on the total pond albedo, and the bare ice and melt pond albedos are combined linearly (Briegleb & Light, 2007):

$$\alpha_{pi}(n) = \frac{h_p(n)}{0.2}\,\alpha_p + \left(1 - \frac{h_p(n)}{0.2}\right)\alpha_i. \tag{1}$$

Because the impact of melt ponds on albedo has been included explicitly, the reduction in bare ice albedo with increasing

25 temperature (Hunke et al., 2015), which was intended to account for melt pond formation, is not included. However, the reduction in snow albedo, $\alpha_s(n)$, with increasing surface skin temperature, intended to take account of the lower albedo of melting snow, has been retained, and takes the form:

$$\alpha_s(n) = \begin{cases} \alpha_c & if\ T(n) < T_c \\ \alpha_c + \left(\frac{\alpha_m - \alpha_c}{T_m - T_c}\right)(T(n) - T_c) & if\ T(n) \geq T_c, \end{cases} \tag{2}$$

where $\alpha_c$ and $\alpha_m$ are the albedos of cold and melting snow respectively, $T_m$ is the snow melting temperature (i.e., 0°C), $T(n)$ is

30 the surface skin temperature of ice in thickness category $n$, and $T_c$ is the threshold temperature, below $T_m$, at which surface melting starts to affect the snow albedo.

**Comment [j2]:** R1-1

The scheme calculates the total gridbox albedo, $\alpha(n)$, of ice in thickness category $n$, for each of the two wavebands by combining the albedo, $\alpha_{pi}(n)$, of the ponded fraction, calculated as described here, with the albedos of bare ice, $\alpha_i$, and snow, $\alpha_s(n)$, weighted by the melt pond fraction, $f_p(n)$, and the snow fraction, $f_s(n)$:

$$\alpha(n) = f_p(n)\alpha_{pi}(n) + \big(1 - f_p(n)\big)\big(f_s(n)\alpha_s(n) + (1 - f_s(n))\alpha_i\big). \tag{3}$$

5   The snow fraction, $f_s(n)$ representing surface inhomogeneity due to windblown snow, for category $n$, is empirically parameterised via a calculation based on snow depth, $h_s(n)$:

$$f_s(n) = \frac{h_s(n)}{h_s(n) + h_{snowpatch}}, \tag{4}$$

where $h_{snowpatch}$ is a length scale parameter (Hunke et al. 2015). Note that this is different from the parameterisation used in the previous configuration, GSI6.0, described by Rae et al. (2015).

10   **2.2. Thermodynamics**

GSI8.1 is the first sea ice configuration of the Met Office model to use multi-layer thermodynamics. Previously, the sea ice model used the zero-layer formulation described in the appendix to Semtner (1976), in which surface temperature reacts instantaneously to surface forcing, and conduction within the ice is uniform. In the new formulation, the sea ice has a heat capacity, which depends on the temperature and salinity and hence conduction can vary in the vertical. The ice is divided

15   into four vertical layers, each with its own temperature and prescribed salinity (a fixed salinity profile); an additional snow   [Comment [j3]: R1-3]

layer is permissible on top of the ice (Figure 1). The thermodynamics scheme is very similar to that described by Bitz and Lipscomb (1999), present in CICE5.1.2, in which the diffusion equation with temperature-dependent coefficients is solved by the iteration of a tridiagonal matrix equation. However, it is modified as described by West et al (2016), with surface exchange calculations, carried out, separately for each thickness category, in the Met Office surface exchange scheme,   [Comment [j4]: R2-0]

20   JULES. The use of JULES allows, allowing near surface temperature to evolve smoothly on the atmosphere time-step (West et al., 2016) which is short compared with the atmosphere-ocean coupling frequency. The modular structure of JULES   [Comment [j5]: R1-4]

allows a consistent treatment of surface exchange (vegetation canopies, snow, soils and sea ice) throughout the model (Best et al., 2011), with the sea ice fraction treated in the same manner as the subgrid tiling of land surface type. All parameters passed between JULES and CICE at each coupling step are shown in the table in Appendix C. The diffusion equation is

25   forced from above by the conductive flux from the ice surface into the top layer interior, which for each ITC is calculated by the surface exchange and passed to the ice model. The category top layer temperature, thickness and conductivity then become the bottom boundary conditions for the next iteration of the surface exchange.

[Figure]

**Figure 1. Schematic demonstrating the time evolution of ice temperature, following an increase in downward short wave surface flux, for: (a) the GSI6 zero-layer ice thermodynamics scheme; (b) the GSI8 multilayer scheme.**

**2.2.1 Semi-implicit coupling**

The OASIS-MCT coupler used within GC3.1 does not have the functionality to regrid variables with time-varying weights. Consequently, for the atmosphere-ice coupling the conductive flux $F_{condtop}$, (on ITC) calculated by the surface exchange for a single atmosphere (low resolution) grid cell, is divided amongst the underlying ocean model (high resolution) cells in proportion to grid cell area. This means that ocean cells with a low ice fraction receive too much energy, while cells with a high ice fraction receive too little. The problems resulting are twofold: an unphysical 'inverse imprint' of ice fraction occurs in the spatial pattern of conductive flux (as shown in Figure 2e), and in a large number of instances the CICE temperature solver is forced with exceptionally high local conductive fluxes - resulting in the iterative temperature solver failing to converge.

In order to render the coupling more physically realistic, and thereby increase the reliability of thermodynamic convergence, the coupling was made semi-implicit. The sea ice fraction is now passed by first-order conservative regridding to the atmosphere at a coupling instant, and this new sea ice fraction used within JULES to apportion $F_{condtop}$ to produce a 'pseudo-local' conductive flux. This new flux is then passed to the ocean model in the normal way, where it is multiplied by the ice fraction field on the ocean grid to produce the grid-box-mean field that is implemented over the ensuing time step

(Figure 2). The grid-box-mean field has the favourable properties of both conserving energy, and of restricting incoming atmospheric fluxes to be proportional to the ocean grid underlying ice fraction which improves convergence of the CICE temperature solver (see Figure 2f). Coupled fields can be shown to be exactly equivalent to the physically desirable solution that would be produced if fluxes were divided amongst underlying ocean grid cells in proportion to ice area. The semi-implicit coupling was found to conserve energy to a similar order of magnitude to the previous, explicit coupling, with an average grid cell error of under $10^{-4}$ Wm$^{-2}$ across the Arctic.

**Comment [j6]:** R1-5

[Figure]

[Figure]

**Figure 2. Demonstrating the implementation, and effect, of semi-implicit coupling within the CICE-JULES surface exchange scheme.. (a) shows an example atmospheric grid cell, overlying 4 ice/ocean grid cells calculates a sea ice conductive flux representing 210J energy; (b) the division of energy between cells with standard coupling, in proportion to grid cell area; (c) the division with semi-implicit coupling, in proportion to ice area. (d) shows a conductive flux field on the atmosphere grid; the resulting flux field on the sea ice grid is shown for (e) standard coupling and (f) semi-implicit coupling.**

With implementation of the semi-implicit coupling there remained two cases in which the CICE temperature solver, forced by the JULES conductive flux, would fail to converge. In the first case, convergence becomes very slow for thin (<0.2m), melting ice, the surface exchange scheme would occasionally calculate large conductive fluxes for which the solver failed to converge within the required 100 iterations. To deal with this problem, a maximum threshold of $1000h_I$ W m-3 (where h$_I$ is

ice thickness in metres) is specified for the conductive flux; any surplus conductive flux above this value is repartitioned to the base of the ice, and added to the ice-to-ocean heat flux. The second issue is a consequence of the way in which the CICE thickness distribution interacts with the coupling method. At high latitudes it is common for a large number of ocean cells each to underlie partially a single atmospheric grid cell. In cold winter conditions, conductive to strong ice growth, the fraction of ice in the thinnest ITC,, $a_1$, cancan be very small. With cold atmospheric temperatures, surface flux and conduction through ice in this category are necessarily strongly upwards; in some cells, random effects, perhaps dynamical, would cause conduction to be stronger than in others, and also lead to lower top layer temperatures. However, stronger conduction also promotes ice growth, which reduces the fraction $a_1$ in the grid cell, as it gets promoted to ITC $a_2$, rendering its top layer temperature less visible to the atmosphere. In a small number of cases this was found to cause runaway cooling, with top layer temperature in isolated cells cooling to below -100°C, forced by high negative conductive fluxes calculated by a surface exchange scheme that was seeing much higher gridbox mean ice temperatures. This problem was solved by linearly reducing conductive flux to zero as top layer ice temperature fell from -60°C to -100°C, with the excess flux passed directly to the bottom of the ice (and therefore helping to grow more ice, the effect that would be expected to occur in reality). These two processes were found to direct an average of 0.4Wm$^{-2}$ and -0.2 Wm$^{-2}$ respectively to the ice base over the course of a year in the Arctic.

**Comment [j7]:** R1-5

**2.3. Dynamics**

The standard elastic-viscous-plastic rheology (EVP) for ice dynamics in CICE is used here (Hunke et al., 2015). However, NEMO is on a C-grid and CICE on a B-grid. This is dealt with though simple interpolation from CICE to NEMO as described in Hewitt et al. (2011). The remapping is the transport/advection algorithm scheme and ridging schemes which are the default in CICE.

**3. Model evaluation**

An example of the sea ice evaluation provided here is the Arctic multiannual mean winter (December, January and February, DJF) ice thickness as diagnosed by the model (Figure 3), both in its CMIP6 configuration of GC3.1 and its previous stable version GC2 (Williams et al., 2015). The model present-day control is forced by greenhouse gases and aerosols from year 2000 for 100 simulated years. The evaluation data is Cryosat-2 satellite thickness (Tilling et al., 2016) inferred from freeboard measurements from 2011-2015 along with the 1990-2010 mean thickness from Pan-Arctic Ice Ocean Modelling and Assimilation System sea ice reanalysis (PIOMAS; Zhang & Rothrock, 2003; Schweiger et al., 2011), inferred through the assimilation of observed sea ice fraction. The Cryosat-2 thickness retrievals are included solely as a guide to the ice thickness distribution as the ice has thinned since 2000. Figure 3 also show the model sea ice extent, depicted by the 15% ice concentration contour, compared with the 1990-2009 mean from the HadISST1.2 sea ice analysis (Rayner et al., 2003). The PIOMAS and GC3.1 Arctic ice thicknesses are comparable in spatial pattern save for PIOMAS depicting a larger area of

thick ice adjacent to North Greenland and the Canadian Archipelago and GC3.1 depicting thicker ice in the central Arctic. CryoSat-2 depicts thinner ice in the western Arctic. Both GC3.1 and CryoSat-2 show thicker ice along the east Greenland coast than PIOMAS. The DJF ice extent in GC2 is low compared with the PIOMAS analysis and CryoSat-2 data, but consistent with the low ice thickness. The extent compares well with the HadISST analysis in GC3.1, however, the ice is

5    overly extensive in the Greenland and Norwegian Seas... This is likely because the deep Atlantic water, in the ORCA025 configuration, is predominately formed in the Labrador Sea and there is very little convection, contrary to observations (Pickart et al., 2003), in the Greenland and Iminger Seas. As a consequence the waters off the East Greenland are rather static and the surface waters cool resulting in excess sea ice. The winter sea ice extent simulated by GC3.1 is much closer to the HadISST observations than was the case for GC2 in the Bering/Chukchi, Barents and Labrador Seas. The Antarctic sea

10   ice extent has improved considerably between GC2 and GC3.1 (Figure 4), the difference being due to a substantial, although not complete, reduction in the Southern Ocean warm bias (see below).

Figure 5 shows the mean seasonal cycle of volume for the GC3.1 model compared with that from GC2. The veracity of the model seasonal cycle of sea ice volume informs us if the annual energy budget to the ice is well balanced. Unfortunately there are few observational means to assess this and so here we use the PIOMAS model as a reference in the Arctic, and

15   satellite estimates from ICESat for the Antarctic (Kurtz & Markus, 2012). It can be seen that, in agreement with Figure 3, the GC2 Arctic ice volume is low, and that it is in closer agreement with PIOMAS at GC3.1. Both models have near identical annual cycles suggesting that, under present day forcing, the new sea ice physics has not substantially altered the seasonal energy balance. The estimates for 2003-2008 Antarctic ice volume from ICESat are for a minimum of 3357 km$^3$ in summer to a maximum of 11,111 km$^3$ in winter (Figure 5). The GC3.1 volume compares well with ICESat in the summer (2735 km$^3$)

20   but is a little higher in winter (17,087 km$^3$). The GC2 configuration had a warm bias in the Southern Ocean, principally caused by excess solar insulation due to low cloud reflectivity (Williams et al., in press), which was melting the Antarctic ice. This bias has been considerably reduced in CG3.1 (Hyder et al., under review) resulting in a substantial increase in the Antarctic sea ice volume.

[Figure]

**Figure 3. Mean winter (December, January and February) Arctic sea ice thickness from the** HadGEM3-GC3.1 (50 year mean from year 2000 equilibrium simulation) (right), the PIOMAS (1990-2009) model reanalysis (upper-left), and inferred from the CryoSat-2 (2011-2015) sea ice freeboard measurements (lower-left). The orange and black lines show the 15% ice concentration contours for the model simulations and the HadISST1.2 sea ice analysis respectively.

[Figure]

**Figure 4. The Antarctic winter (June, July & August) mean ice extent with a) HadGEM3-GC2 and b) HadGEM3-GC3.1 (black) compared with the HadISST1.2 sea ice analysis (orange)**

[Figure]

**Figure 5. The HadGEM3 model annual cycle of sea ice volume in the Arctic (upper) and Antarctic (lower). Volume estimates from the PIOMAS model reanalysis are included for the Arctic and ICESat estimates of volume in the Antarctic (grey dashed lines).**

**4. Summary**

The GSI8.1 sea ice configuration of the Met Office Hadley Centre CMIP6 coupled model HadGEM3-GC3.1 has a number of physical enhancements compared to the previous version GSI6, including the introduction of multilayer thermodynamics and an explicit representation of the radiative impact of meltponds. A semi-implicit coupling scheme refines the transpose of atmospheric fluxes to the sea ice, improving the stability of the thermodynamic solver. The final GC3.1 namelist options and pre-processor keys (see Appendix A and Appendix B) produce ice thickness and extent that are in good agreement with analyses.

**5. Code availability**

Due to intellectual property right restrictions, we cannot provide either the source code or documentation papers for the UM or JULES. The Appendices to this paper does include a set of Fortran namelists that define the configurations in the coupled climate simulations

The Met Office Unified Model (MetUM) is available for use under licence. A number of research organisations and national meteorological services use the UM in collaboration with the Met Office to undertake basic atmospheric process research, produce forecasts, develop the UM code and build and evaluate Earth system models. For further information on how to apply for a licence see http://www.metoffice.gov.uk/research/modelling-systems/unified-model.

JULES is available under licence free of charge. Further information on how to gain permission to use JULES for research purposes can be found at https://jules.jchmr.org/software-and-documentation.

The model code for NEMO v3.6 is available from the NEMO website (http://www.nemo-ocean.eu). On registering, individuals can access the code using the open-source subversion software (http://subversion.apache.org/).

The model code for CICE is available from the Met Office code repository https://code.metoffice.gov.uk/trac/cice/browser

In order to implement the scientific configuration of GC3.1 and to allow the components to work together, a number of branches (code changes) are applied to the above codes. Please contact the authors for more information on these branches and how to obtain them.

**6. Data availability**

Due to the size of the model data sets needed for the analysis, they require large storage space of order 1 TB. They can be shared via the STFC-CEDA platform by contacting the authors.

**Acknowledgements**

[revised manuscript text omitted]

Appendix C. Variables passed between CICE and JULES through the OASIS coupler

| From CICE to JULES | From JULES to CICE |
|---|---|
| Ice thickness (per category) (m) | X component of wind stress (grid-box mean) ($Nm^{-2}$) |
| Ice area fraction (per category) | Y component of wind stress (grid-box mean) ($Nm^{-2}$) |
| Snow thickness (per category) (m) | Rainfall rate (grid-box-mean) ($kgm^{-2}s^{-1}$) |
| Top layer ice temperature (per category) (K) | Snowfall rate (grid-box-mean) ($kgm^{-2}s^{-1}$) |
| Top layer effective conductivity (per category) ($Wm^{-2}K^{-1}$) | Ice sublimation * (per category) ($Wm^{-2}$) |
| Melt-pond fraction (per category) | Ice top melting * (per category) ($Wm^{-2}$) |
| Melt-pond depth (per category) (m) | Ice conductive flux * (per category) ($Wm^{-2}$) |
| X component of sea ice velocity (grid-box mean) ($ms^{-1}$) | Ice surface skin temperature (per category) (K) |
| Y component of sea ice velocity (grid-box mean) ($ms^{-1}$) | |

* Indicates fields subject to semi-implicit coupling

**Responses to Referees**

We thank the referees for their helpful comments to prepare a more complete manuscript. We have included comments in the text to show where reviewer major comments have been addressed in the text- (numbered as described below). Some comments have been addressed through an explanation of choices made in the comments following the reviewer's point. The reviewer comments are included below in black and author responses in red.

**Anonymous Referee #1**

The manuscript describes the sea-ice component GSI8.1 of the Met Office coupled configuration HadGEM3-GC3 and its evaluation against PIOMASS reanalysis and Cryosat satellite measurements. This will certainly be a reference paper for CMIP6 simulations and therefore I recommend its publication but with some substantial modifications.

General comments:

**R1-1**. It is very unclear what are the novelties in this version of the sea-ice component (GSI8) compared to the one used for CMIP5 (GSI6). I recommend to explain it in the introduction or at the beginning of section 2.

Added a description of main advances between GSI6 and GSI8.1 in section 2 – model description.

GSI6 was the version used in HadGEM3-GC2 which is not the model used in CMIP5 (which was HadGEM2-ES). The difference between CMIP5 (HadGEM2-ES) and CMIP6 (HadGEM3) are comprehensive with a new ocean model (NEMO), a new atmospheric dynamic core(ENDGAME), new aerosols (GLOMAP) and a complete overhaul of the land surface scheme (JULES). With all these changes we do not compare the differences in sea ice between CMIP5 and CMIP6, but rather the advances since the last stable version of the HadGEM3 model (GC2 the description of which is found in Rea et al.(2015))

**R1-2**. It would be valuable to know why the most recent developments in CICE are not included in GSI8, especially those which have been available for a couple of years. I am thinking of Delta-Eddington scheme, more elaborate melt ponds (which maybe necessitates the use of D-E), EAP rheology, mushy layers etc. Is it a question of timing, performance, robustness (testing)?

We do not consider it appropriate to describe what is **not** included in the model, but rather what **is** included in the model. Since our coupling to the ice is through the surface scheme (JULES), we do not use the D-E because it is not consistent with the rest of the surface scheme. We are developing our own two stream radiation model for snow and ice (similar to D-E) which will be consistent across all model components (vegetation canopy, snow and sea ice). We do of course have a prognostic melt pond scheme from CICE as described in section 2.1. The comprehensive evaluation and tuning of EAP was simply a mater of time available and will likely be switched on in the next version of GSI.

**R1-3**. Besides novelties that need to be explained in details, I think the authors should not forget to give information about the rest of the ice code. Is there an interactive salinity, a fixed profile or a constant salinity? What scheme is used for advection (remapping)? And ridging/rafting? Do they consider biogeochemistry in the ice?

> **Comment [j8]:** R1-2

Section 2.2 states that the salinity is prescribed on thermodynamic layers, but have expanded this to explicitly state use of a fixed salinity profile. We have also included some reference to other components of the model, however, we use the CICE code and advection (remapping) and ridging are as standard in that model. We have now specified in the introduction that the sea ice component, described here, is associated with the physical coupled model and so contains no biogeochemistry. Such will be included in the Earth System Model, UKESM1, but that scheme is not sufficiently complex (computational cost considerations) to warrant including sea ice biology.

**R1-4)** Description of JULES coupling lacks clarity but I come back to it in the "specific comments" section.

See answer below specific comments below

**R1-5)** Did the authors evaluate conservation of mass, salt and heat in the system? What are the leaks (if any) and where do they come from?

We have added numbers on energy conservation at appropriate points in the text, although we do not have figures for the salt conservation – which is normally assessed in the ocean model (NEMO). We find no systematic drift in the salinity budget during long control simulatons. The conservation of heat energy by the new semi-implicit coupling was evaluated. Energy was found to be conserved to within $10^{-4}$ Wm$^{-2}$ over the Arctic, to a very similar order of magnitude to the previous coupling method used in GSI6, in which no implicit weighting by ice area was carried out. The CICE thermodynamic solver is designed to conserve energy to within $10^{-5}$ Wm$^{-2}$ in each grid cell.

**R1-6)** What is the performance of GC3 compared to GC2 in terms of CPU since new physics have been added? How much slower it is?

We do not routinely produce the sea ice code timing diagnostics, and due to other substantial global model changes between GC2 and GC3, the overall timing statistics would not be relevant to this paper. The 90% of the sea ice computational cost is in the Dynamics, and the changes described here are in the Thermodynamics. The sea ice component is ~6% of the total model run time.

> **Comment [j9]:** R1-6

Specific comments:

— 2.1 albedo —

l.6: I do not understand why penetration of radiation is not included. Is it related to JULES? If yes, the authors should say it.

The penetration of SW radiation would require a modification to internal heating rates of thermodynamic layers. JULES is a component of the problem, as regards backscattered radiation, but the issue was not addressed due to time limits. Time limitations are not a description of the model and thus no further comment has been added to the text.

l. 12-15: I understand that the impact of melt ponds on albedo is linear with pond depth in the range [4 mm - 20 cm]. I would have thought more about an exponential dependency as in Lecomte et al. (2011, 2015). Is there a reference for the linear dependency?

The linear dependence comes from the CICE Delta-Edington documentation Figure 13 which shows an exponential fit on a log(depth) plot = linear dependency on depth. Reference is now included in the text

.l. 23: I suppose the value of Tc is -1degC as in Hunke et al. (2015), then refers to dt_snow_cice in the namelist? Maybe it should be stated more clearly. More generally it is sometimes difficult to relate the namelist variables with the text.

Referring to the namelist variables in the text proved clumsy and reduced readability. Where relevant, we have added equation numbers and expanded the namelist (Appendix A) description ofnow specified how the variables to refer to the
5    generic terms used in the equations. in Section 2.1 relate to the namelist variables listed in Appendix A

.l. 30: What does Hsnowpatch represent? Snow blowing by winds? But then, where does this value (2 cm?) come from?

Schemes of this type are often used to represent snow patchiness due to winds and uneven melt (e.g. Oerlemans, J., and W. H. Knap (1998), with values of Hsnowpatch ranging from 0.01 – 0.05m, although this is not the same as the scheme used here). The value of 0.02m was empirically derived but based on a compromise from a wide range of sources. There is no
10   single reference to use here and a brief comment on its empirical derivation has been included in the text..

— 2.2 Thermodynamics —

I find that the JULES interface is poorly described here. For example, it is said in the text that the ice model passes to JULES the top layer temperature, thickness and conductivity. Well it would be clearer to state all the variables that are exchanged: ice concentration, ice and snow thicknesses, temperature and conductivity of the first layer, concentration and thickness of
15   ponds (I think that's all?)

A table now added, Appendix C, on CICE parameters passed through JULES and vice a versa

Also, the authors should say that with JULES there is no transmitted radiation in and through the ice (from what I understood). What consequences do the authors expect on ice temperature or ocean heat budget?

Ice that is less than 15cm thick will have a portion of the SW radiation  transmitted though to the ocean. This fraction is
20   highest during the late melt season. Under present day simulations this is 0.3% of the Arctic ice pack, averaging to 0.02 $W/m^2$ of incoming solar across the Arctic which causes no surface melt. Thus the impact is small in the present day simulations described in this paper,, however, it will be larger as the ice pack becomes perennial. However, we have not done a full energy budget analysis to reliably quote here.

Moreover, the reader must understand why JULES has been chosen (in just a couple of sentences).
25   JULES is the surface exchange scheme that is applied to all components of the model which involve a second solver (e.g. terrestrial snow, soil levels). This is simply how the model works, it a is better solution than other methods as shown by West et al., 2016). It allows the atmospheric solver to be independent from the coupling interval between the components, by providing a smoothly changing surface temperature. An extra sentence has been added 'However, it is modified as described by West et al (2016), with surface exchange calculations carried out in the Met Office surface exchange scheme, JULES,
30   allowing surface temperature to evolve smoothly on the atmosphere time-step which is shorter than the coupling period.  We have added the following "The use of JULES allows near surface temperature to evolve smoothly on the atmosphere time-step (West et al., 2016) which is short compared with the coupling frequency. The modular structure of JULES allows a consistent treatment of surface exchange (vegetation canopies, snow, soils and sea ice) throughout the model (Best et al., 2011), with the sea ice fraction treated in the same manner as subgrid tiling of land surface type."

l. 6: precise that heat capacity depends on temperature and salinity.

Added : "dependence of temperature and salinity"

l. 8: Fig 1 is not properly described (neither in the text nor in the caption). "Old" means the formulation used in GSI6?

Yes – we should have used 'GSI6' and 'GSI8' in place of 'old' and 'new'. This has been corrected.

— 2.2.1 semi-implicit coupling —

p.4, l. 10-18: this part is kind of unclear. I would suggest to rewrite it in a simpler way like: Problem to solve = a) non convergence of the temperature solver, b) coupling not physical. Solution = sea-ice fraction passed to the atm. which allows a semi-implicit calculation of the splitting of Fcond onto the ocean grid cells. This flux is then multiplied by ice fraction in each cell.

We have re-ordered this section so that it starts with a clearer summary of the problem to be solved

p.5, l. 11: "1000Hi". What does this threshold mean?

In the text it says that this denotes ice thickness. Sorry that this is not clearer – the word order has been altered.

p.5, l. 12: Is there any quantification of this "surplus conductive flux"? I guess there is still a small heat leak due to the non-perfect convergence of the iterative procedure. I think it would be valuable to diagnose it. In Hunke et al. (2015), it is said that this flux is about 0.01 W/m2. Does this value still hold? What does it mean in terms of global heat leak?

We did diagnose the Northern Hemisphere totals of this surplus conductive energy, albeit in a test while we were still using the previous version of the model, CICE 4. We have added sentences at the end of the relevant paragraphs to summarise our findings. The NH averages of surplus flux were small but non-negligible in the context of the overall ice heat budget, 0.4 $Wm^{-2}$ and -0.2 $Wm^{-2}$ respectively. However, this is not a leakage from the model system as the energy is exactly compensated for in the ocean-ice system.

— 3. model evaluation —

p.6 l. 27: "this being thin ice that microwave unable detect". I do not understand this sentence.

Sentence restructured to omit this phrase. A broader discussion of sea ice biases has been included

p.7 l. 3-5: Do the authors mean that changes in Arctic sea-ice can be solely attributed to changes in the sea-ice component (contrary to the Antarctic)? I am not convinced that identical seasonal cycles indicate that forcings are unchanged. The mean forcing could also change while keeping the same annual cycle. Am I wrong?

Rephrased to suggest that it means that the new model physics has not changed the seasonal mass-energy budget.

p.7 l. 9: Concerning the great improvement in Antarctic sea ice (at least in summer), I understand it is solely due to the reduction of the warm bias in the ocean but does the atmosphere play any role here?

Excess solar insulation, usually due to poor representation of mixed phase clouds, is the reason for southern ocean warm bias in most models (plus some turbulent heat issues). Now stated explicitly.

— References —

Lecomte, O., Fichefet, T., Vancoppenolle, M., & Nicolaus, M. (2011). A new snow
thermodynamic scheme for large-scale sea-ice models. Annals of Glaciology, 52(57),

337-346.

Lecomte, O., Fichefet, T., Flocco, D., Schroeder, D., & Vancoppenolle, M. (2015). Interactions between wind-blown snow redistribution and melt ponds in a coupled ocean–

sea ice model. Ocean Modelling, 87,

**Anonymous Referee #2**

The authors describe the new sea ice model component of the climate model HadGEM3-GC3.1. As this model will be

applied in a large number of experiments, this description is an important source of information for the community and will

10    be very helpful for the users of the outputs of HadGEM3-GC3.1. The paper is short but focused on the important points.

However, I would personally have been interested in a comparison of the choices performed here with the ones made in

other state-of-the-art models, not just in the previous or other versions of the same model. This would have provided a wider

perspective of the paper.

15    It would be difficult to assess the sea ice components of other CMIP6 models with out a comprehensive review of the

associated documentation, which for many models does not yet exist

R2-0 For me, the only point that absolutely requires additional discussion is the coupling and how it interacts with the ice

thickness distribution scheme.  If I understand well (page 3, lines 11-15), the surface temperature of the sea ice or snow and

20    surface melt are computed by the land surface module that transfers to the sea ice model a conduction flux which is used as

boundary condition for the sea-ice model.  This is described in West et al. (2016) but a few additional explanations would be

helpful here in the revised version. The distribution of the conductive flux between the grid cells is described in section

2.2.1. However, I was not able to understand from the information given in the manuscript the method applied to distribute

this conductive flux among ice thickness categories. It is also not clear to me if the surface melt is the same for all the

25    categories. The thickness is mentioned page 5, lines 11-13 but it is to solve a specific problem of instability, not related to

the ice thickness distribution. It is expected that the melting and conduction fluxes are very different between the different

categories as they should be strongly related to the thickness. If in the proposed implementation, the same conductive flux is

given for all categories, my interpretation is that nearly all the interest of the ice thickness distribution scheme is lost. If this

is the case, this should be clearly mentioned as a strong limitation of the approach. If I have misunderstood something, the

30    authors should explain their arguments justifying their choices and how this takes advantage of the ice thickness distribution

scheme.

The reviewer is absolutely correct that this is not clear in the text. Of course the surface exchange scheme, and the

atmosphere-ice coupling,perform separate computations for each ice thickness category. This enables conductive flux to be

realistically simulated across the ice thickness distribution in a similar manner to that which would occur in the standard

CICE setup, with surface exchange calculated in the ice thermodynamics solver. We have added a sentence in the opening paragraph of Section 2 to explain this, We have also modified the first paragraph of the 'Thermodynamics' section, to explain that all calculations described occur separately for each thickness category.

I have also some very minor remarks.

1/ Does the atmospheric component have a specific name (as the ocean or surface ones) ?

Yes thanks for spotting this – the 'Met Office Unified Model' this has now been added to the model description

2/ Page 11, line 11. This would be useful to list the 'GC3.1 ocean-atmosphere variables'.

The coupling variables between sea ice and atmosphere are now listed in Appendix C

3/ Page 6, lines 28-29. Here and for the other points mentioned in the manuscript, this would be interesting to discuss briefly the causes of the differences of the results of GC2 and GC3.1.

A description of the general changes between GC2 and GC3.1 is now given at the beginning of the model description these changes result in a different climate forcing to the sea ice, mainly due to the clouds and aerosol components. Some commentary on the northern sub-polar seas is also added.

4/ Page 7, lines 4-5. I do not understand why similar annual cycles but with an offset indicate similar oceanic and atmospheric forcing. I guess a change in oceanic or atmospheric forcing could also be the origin of the offset.

This has now been rephrased to indicate that the ice physics changes between GC2 and GC3.1 have not resulted in a change to the seasonal ice budget.

5/ Page 7, line 5. A comparison with the observed ice extent in the Southern Ocean should also be provided. Figure 3. The months selected should be given in the figure.

Added the months in figure 3. We have also added a new figure (fig 4) to show the Southern Hemisphere winter ice extent verification.